# Metformin Is Associated with the Inhibition of Renal Artery AT1R/ET-1/iNOS Axis in a Rat Model of Diabetic Nephropathy with Suppression of Inflammation and Oxidative Stress and Kidney Injury

**DOI:** 10.3390/biomedicines10071644

**Published:** 2022-07-08

**Authors:** Amal F. Dawood, Amro Maarouf, Norah M. Alzamil, Maha A. Momenah, Ayed A. Shati, Nervana M. Bayoumy, Samaa S. Kamar, Mohamed A. Haidara, Asmaa M. ShamsEldeen, Hanaa Z. Yassin, Peter W. Hewett, Bahjat Al-Ani

**Affiliations:** 1Department of Basic Medical Sciences, College of Medicine, Princess Nourah Bint Abdulrahman University, Riyadh 11671, Saudi Arabia; afdawood@pnu.edu.sa; 2Department of Physiology, Kasr Al-Aini Faculty of Medicine, Cairo University, Cairo 12613, Egypt; haidaram@cu.edu.eg (M.A.H.); asmaa.abdulwahab@kasralainy.edu.eg (A.M.S.); hanaa.z.yassin@kasralainy.edu.eg (H.Z.Y.); 3Department of Clinical Biochemistry, Birmingham Heartlands Hospital, University Hospitals Birmingham NHS Foundation Trust, Birmingham B9 5SS, UK; a.maarouf@nhs.net; 4Department of Clinical Science, Family Medicine, College of Medicine, Princess Nourah Bint Abdulrahman University, Riyadh 11671, Saudi Arabia; nmalzamil@pnu.edu.sa; 5Department of Biology, College of Science, Princess Nourah Bint Abdulrahman University, Riyadh 11671, Saudi Arabia; mamomenah@pnu.edu.sa; 6Department of Child Health, College of Medicine, King Khalid University, Abha 62529, Saudi Arabia; ashati@kku.edu.sa; 7Department of Physiology, College of Medicine, King Saud University, Riyadh 11461, Saudi Arabia; nbayoumy@ksu.edu.sa; 8Department of Histology, Kasr Al-Aini Faculty of Medicine, Cairo University, Cairo 12613, Egypt; samaw@kasralainy.edu.eg; 9Institute of Cardiovascular Sciences, College of Medicine and Dental Sciences, University of Birmingham, Birmingham B15 2TT, UK; p.w.hewett@bham.ac.uk; 10Department of Physiology, College of Medicine, King Khalid University, Abha 62529, Saudi Arabia

**Keywords:** diabetic nephropathy, AT1R/ET-1/iNOS axis, renal artery, mesangial expansion, kidney fibrosis, metformin

## Abstract

Diabetes is the most common cause of end-stage renal disease, also called kidney failure. The link between the renal artery receptor angiotensin II type I (AT1R) and endothelin-1 (ET-1), involved in vasoconstriction, oxidative stress, inflammation and kidney fibrosis (collagen) in diabetes-induced nephropathy with and without metformin incorporation has not been previously studied. Diabetes (type 2) was induced in rats and another group started metformin (200 mg/kg) treatment 2 weeks prior to the induction of diabetes and continued on metformin until being culled at week 12. Diabetes significantly (*p* < 0.0001) modulated renal artery tissue levels of AT1R, ET-1, inducible nitric oxide synthase (iNOS), endothelial NOS (eNOS), and the advanced glycation end products that were protected by metformin. In addition, diabetes-induced inflammation, oxidative stress, hypertension, ketonuria, mesangial matrix expansion, and kidney collagen were significantly reduced by metformin. A significant correlation between the AT1R/ET-1/iNOS axis, inflammation, fibrosis and glycemia was observed. Thus, diabetes is associated with the augmentation of the renal artery AT1R/ET-1/iNOS axis as well as renal injury and hypertension while being protected by metformin.

## 1. Introduction

Diabetic nephropathy (DN, also known as diabetic kidney disease) is the most common cause of renal failure. It represents over 40% of admitted cases of end-stage renal disease in the USA, and about one-third of patients living with diabetes have DN [1,2]. The type of obesity that induces insulin resistance is involved in the development of diseases such as diabetes, hypertension, fatty liver, and cardiovascular diseases, which are well recognized as hallmarks of metabolic syndrome [3,4]. Excessive consumption of a Western diet rich in processed food and decreased physical activity account for the rapid rise in obesity in genetically vulnerable people and cause non-communicable diseases such as diabetes, which claims the lives of millions around the world [3,4]. Additionally, obesity is also associated with renal dysfunction and contributes to DN, glomerulopathy and proteinuria and end-stage renal disease [5,6,7].

Kidney injury due to diabetes affects glomerular filtration barrier cells such as podocytes and glomerular endothelial cells and causes mesangial expansion and thickening of the glomerular basement membrane due to deposition of collagen and other proteins and leads to proteinuria [8,9]. This, together with the activation of the renin–angiotensin system (RAS), which increases the glomerular capillary pressure and decreases the glomerular filtration rate (GFR), contribute to the development of DN [8,9]. Indeed, (i) in patients with DN, the inhibition of RAS with captopril, an angiotensin-converting enzyme inhibitor, reduced the risk of DN patients reaching end-stage renal disease by half [9]; (ii) proteinuria is a recognized marker of kidney injury and is used to monitor the prognosis and the response to treatment of injured kidneys [10]; and angiotensin II type I receptor (AT1R) inhibitors decreased proteinuria in patients with DN [11].

Damaged renal blood vessels in DN tip the balance towards vasoconstriction and hence blood pressure elevation and endothelial dysfunction. This plays an important role in the prognosis of the disease via the augmentation of endothelin (ET-1) release [12] as well as inhibition of eNOS activity, which increases oxidative stress [13]. The antidiabetic drug metformin was reported to treat renal diseases such as DN by decreasing renal inflammation, oxidative stress and fibrosis in animals and to reduce the progression to end-stage renal disease in humans [14]. Metformin also protects against acute kidney injury induced by gentamycin in rats [15]. However, the association between the renal artery AT1R/ET-1/iNOS axis and DN with and without metformin administration has not been previously investigated. Thus, our aim in this study was to investigate whether such an association can be demonstrated as well as whether the anti-diabetic drug metformin was able to ameliorate any potential deleterious findings.

## 2. Materials and Methods

### 2.1. Animals

All experimental protocols (protocol number H-01-R-059) were authorized by the research ethical committee at Princess Nourah Bint Abdulrahman University in accordance with the Guide for the Care and Use of Laboratory Animals published by the US National Institutes of Health (NIH publication No. 85-23, revised 1996). The experiments were carried out on Wistar male rats weighing 170–200 g. During the acclimation period, rats were fed standard pellets. The rats were also given free access to water and kept in a clean environment with a 12 h light/dark cycle.

### 2.2. Experimental Design

After a one-week adaptation period, 24 rats were randomly allocated into three groups: control group (Control), comprised of non-diabetic, non-treated rats fed with standard laboratory chow for 12 weeks; and diabetic type 2 group (T2DM), in which diabetes mellitus was induced in this group of rats using standard methods, i.e., a combination of a high carbohydrate and fat diet (HCFD) and a single injection of streptozotocin, as previously described [16]. These rats were kept on a HCFD until the end of week 12. The metformin plus T2DM group (Met + T2DM) consisted of animals that were given metformin (200 mg/kg) from day 1 and were fed HCFD for 2 weeks before T2DM was induced. For another 10 weeks, the rats were given metformin and were fed an HCFD. Blood samples were collected by heart puncture under anesthesia (sodium thiopentone at 40 mg/kg body weight) at the end of experiment; thereafter, rats were euthanized by cervical dislocation and cardiac tissue samples were taken. Diabetes was confirmed in the model group 1 week after STZ injection using a Randox reagent kit to measure fasting blood glucose levels (>200 mg/dL) (Randox Laboratories Ltd., Crumlin, UK ).

### 2.3. Determination of Glucose, HbA1c, Urea, Creatinine, High Sensitivity C-Reactive Protein (hs-CRP), Tumor Necrosis Factor-Alpha (TNF-α), Interleukin-6 (IL-6), Malondialdehyde (MDA), Superoxide Dismutase (SOD), Glutathione Peroxidase (GPx), Urine Albumin and Ketones

A Randox reagent kit was used to colorimetrically determine glucose levels in the blood (Randox Laboratories Ltd., Crumlin, UK). An ELISA kit (Cat. # 80300; Crystal Chem, Inc., Elk Grove Village, IL, USA) was used to assess HbA1c levels in the blood. Colorimetric methods were used to test blood urea and creatinine, as directed by the manufacturer (BioAssay Systems, Hayward, CA, USA). ELISA kits were used to test hs-CRP (Cat. # ERC1021-1, ASSAYPRO, St Charles, MO, USA), TNF-α (Biotang Inc., Lexington, MA, USA), IL-6 (Biotang Inc., Lexington, MA, USA), MDA (Cayman Chemical, Ann Arbor, MI, USA), GPx (Cayman Chemical, Ann Arbor, MI, USA). ELISA kit was used to measure urinary albumin levels (Novus Biologicais, Centennial, CO, USA). Urine dipsticks were used to test for ketonuria and specific gravity (Williams Medical Supplies, Gwent, UK).

### 2.4. Quantitative Real-Time Polymerase Chain Reaction (qRT-PCR) of iNOS and eNOS Gene Levels

qRT-PCR was carried out as explained previously [17]. To summarize, total RNA was isolated from rat renal arteries using Trizol Reagent (Qiagen, Germantown, MD, USA) and reverse-transcribed using the Fermentas cDNA synthesis kit (Fermentas, Waltham, MA, USA) according to the manufacturer’s instructions. cDNA samples were amplified with primers specific for iNOS (sense 5′-CACCACCCTCCTTGTTCAAC-3′ and antisense 5′-CAATCCACAACTCGCTCCAA-3), eNOS (sense 5′-TATTTGATGCTCGGGACTGC-3′ and antisense 5′-AAGATTGCCTCGGTTTGTTG-3′) and β-actin in Master Mix containing SYBR-Green Supermix (Molecular Probe, Eugene, OR, USA). The comparative Ct technique was used to calculate the relative gene expression levels.

### 2.5. Western Blotting Analysis of AT1R

Proteins were collected from renal arteries and immunoblotted using 25 μg of protein per sample, as described earlier [17]. Anti-angiotensin II Type-1 receptor polyclonal antibody (AT1R, Catalog # ABIN6288600, antibodies-online GmbH, Aachen, Germany) was used to probe membranes overnight at 4 °C. The ECL detection kit was used to visualize the proteins (Merck Life Science, Gillingham, Dorset, UK). On the Chemi Doc MP imager, relative expression was calculated by reading the band intensity of the target proteins against the control sample using Image analysis software after normalization by β-actin.

### 2.6. Immunostaining and Assessment of Kidney Pathology

Kidney specimens were obtained from various experimental groups and fixed in 10% formal saline for 1 day before dehydration with ascending grades of alcohols, followed by clearing and embedding these samples in paraffin using conventional techniques, as previously reported [18]. Paraffin blocks were sectioned into 5 μm thick slices and stained with Periodic acid Schiff (PAS) as well as immunostained overnight at 4 °C with anti-iNOS (Cat # ab15323, Abcam, Cambridge, UK) antibody, followed by half an hour incubation with a secondary antibody at room temperature. Meyer’s hematoxylin was used to counterstain the sections.

To measure renal collagen deposition, we used Sirius red staining of tissue sections. Embedded kidney slices were dewaxed in xylene and rehydrated in water via a series of graded alcohols. After an overnight incubation with 0.1 percent Sirius red (Sigma-Aldrich, Gillingham, Dorest, UK), the slides were dipped in 0.01 M hydrochloric acid and dehydrated with progressive ethanol doses without water. Morphometry of the areas’ percent collagen deposition in Sirius Red stained sections and the areas’ percent iNOS immunostaining were done in 10 non-overlapping fields for each group using a “Leica Qwin 500 C” image analyzer (Cambridge, UK). The means and standard deviations (SD) of quantitative data were calculated and compared using analysis of variance (ANOVA) and post hoc analysis (Tukey test). *p*-values less than 0.05 were deemed statistically significant. The calculations were performed using Version 19 of the statistical package for social science (SPSS) software.

### 2.7. Determination of Tissue Levels of ET-1 and Advanced Glycation End Products (AGEs)

Renal artery tissue samples from all groups of rats were washed in phosphate buffered saline (PBS) at pH 7.4. They were then homogenized in a cold phosphate buffer containing ethylene-diamine-tetra-acetic acid using an ultrasonic homogenizer (EDTA). Each rat’s supernatant was aliquoted into separate tubes and stored at –70 °C for analysis of ET-1 and AGEs levels using ELISA kits provided by Abcam, Cambridge, UK, and Ray Biotech, USA, respectively, according to manufacturer’s instructions.

### 2.8. Determination of Mean Arterial Blood Pressure

The tail-cuff technique was used to measure blood pressure in conscious rats (BP monitor, LE 5001, LETICIA scientific Instruments, Barcelona, Spain), as previously reported [18].

### 2.9. Statistical Analysis

The means ± standard deviations were used to represent the data (SD). SPSS version 10.0 was used to analyze the data (SPSS, Inc., Chicago, IL, USA). Tukey’s post hoc test was used after performing a one-way ANOVA. For the purpose of detecting a possible relevance between two separate values, a Pearson correlation statistical analysis was used. If the *p*-value was less than or equal to 0.05, the results were considered significant.

## 3. Results

### 3.1. Metformin Is Associated with the Inhibition of the AT1R/ET-1 Axis and AGEs in the Renal Artery Induced by Diabetes

Over stimulation of AT1R and ET-1R vascular receptors by angiotensin II (AngII) and endothelin-1 (ET-1), respectively, lead to sustained vasoconstriction and hence blood pressure elevation and endothelial dysfunction [19]. In addition, AngII can increase the expression of ET-1 [20] and the activity of ET-1 is increased in patients with type 2 diabetes mellitus (T2DM) [21]. To determine whether the AT1R/ET-1 axis is induced by diabetes in the renal artery of the model group of rats and whether metformin treatment can inhibit this process, we assessed tissue levels of AT1R and ET-1 in all rat groups 10 weeks after the induction of T2DM (Figure 1). Western blot and ELISA analyses showed an increase in the expression of AT1R and ET-1 proteins by diabetes, which appeared to be significantly (*p* ≤ 0.0002) inhibited by metformin (Figure 1A–C). Metformin treatment also significantly (*p* ≤ 0.0013) decreased glycemia (Figure 1D,E) and renal artery tissue levels of AGEs (Figure 1F) compared to the untreated diabetes group.

### 3.2. Metformin Protects against Diabetes-Modulated NOS Enzymes in Renal Artery and Kidney Tissue

eNOS knockout mice exhibit exacerbated diabetic nephropathy [13] and the upregulation of iNOS, causing damage to the proximal renal tubule [22]. We investigated the hypothesis that diabetes augments iNOS and ameliorates eNOS in the renal artery of rats and that metformin can protect against the actions of diabetes. qRT-PCR analysis of renal artery tissue samples prepared from the T2DM group showed an increase in iNOS (Figure 2A,B) and a decrease in eNOS (Figure 2A,C) gene expression. Metformin treatments significantly (*p* ≤ 0.0005) protected against these adverse changes in comparison to the untreated diabetes group.

We then evaluated levels of nitrosative stress (iNOS) induced by diabetes in kidneys harvested from all animal groups and assessed disease phenotype in the presence and absence of metformin. Immunohistochemical staining of iNOS (Figure 2D–F) prepared from kidney sections showed that diabetes enhanced the number of iNOS + ve immunostaining cells (Figure 2E) when compared to cells which stained negative in the control group (Figure 2D). Metformin treatment appeared to markedly decrease iNOS + ve (Figure 2F,G) kidney tissue immunostaining to significant levels compared to the control group of rats.

Based on the above results showing that diabetes upregulated the vasoconstrictive proteins and downregulated the vasorelaxant eNOS gene in renal arteries that are blocked by metformin, we measured the mean arterial blood pressure (MAP) in all rats (Figure 2H). Diabetes significantly (*p* < 0.0001) induced increased MAP, which was inhibited by metformin. However, the effects of metformin were still significant (*p* = 0.0124) when compared with the untreated control group.

### 3.3. Metformin Inhibits Biomarkers of Inflammation and Oxidative Stress Induced by Diabetes

The AT1R antagonist losartan inhibited renal inflammatory and oxidative stress biomarkers as well as reduced blood pressure [23]. Therefore, in view of the upregulation of AT1R in our model of diabetic nephropathy, we evaluated levels of inflammation and oxidative stress markers induced by diabetes in all animal groups. Table 1 shows that diabetes substantially augmented inflammation (hs-CRP, TNF-α and IL-6) and oxidative stress (MDA measured as lipid peroxidation), whereas, levels of antioxidants (SOD and GPx) were significantly inhibited by diabetes. All these parameters were protected by metformin.

### 3.4. Metformin Is Associated with Inhibition of Mesangial Matrix Expansion and Kidney Fibrosis Induced by Diabetes

Mesangial matrix expansion and glomerular basement membrane thickening are hallmarks of diabetic nephropathy that lead to renal fibrosis and eventually to end-stage renal disease [24]. In addition, an AT1 receptor blocker inhibits kidney fibrosis in an animal model of chronic kidney disease [25]. Therefore, in view of the results described above that showed the upregulation of AT1R by diabetes, 10 weeks after the induction of T2DM we assessed the levels of kidney injury and fibrosis in all rat groups (Figure 3). After staining with periodic acid Schiff (PAS), kidneys were examined by light microscopy (Figure 3A–D). Compared to normal kidney architecture (Figure 3A), diabetes caused expanded mesangial regions and thickening of glomerular basement membranes, parietal layers and tubular membranes, with the loss of the brush borders of the tubular cells and PAS-stained regions in the cytoplasm of the tubular epithelium (Figure 3B,C). Metformin treatment substantially protected against diabetes-induced kidney injury (Figure 3D). Furthermore, quantification of mesangial matrix expansion obtained from tissues stained with PAS (Figure 3E) demonstrated an effective (*p* < 0.0001) inhibition of mesangial expansion by metformin to levels comparable to the control group.

Sirius red-stained kidney tissue sections of the model group (T2DM) revealed substantial coarse collagen deposition in the renal interstitium and surrounding blood vessels (Figure 3G) compared with the fine collagen cell staining kidney sections of the control group (Figure 3F). Metformin treatment for 12 weeks significantly (*p* < 0.0001) protected against diabetes-induced collagen deposition (Figure 3H–J), but it was not completely prevented by the drug.

### 3.5. Metformin Inhibits Biomarkers of Kidney Injury Induced by Diabetes

The diabetic kidney produces ketone bodies [26]. We measured biomarkers of kidney injury induced secondary to T2DM 10 weeks post diabetes induction in all rat groups. As shown in Table 2, diabetes caused a sharp increase in proteinuria (urine albumin mg/24 h), ketonuria, urine specific gravity, blood urea and creatinine, which was significantly (*p* ≤ 0.0036) ameliorated by metformin to levels still significant (*p* ≤ 0.0225) to the control group for specific gravity and creatinine. This means a partial inhibition for these parameters by metformin. On the other hand, uncontrolled diabetes significantly (*p* < 0.0001) decreased animal body weight, which was otherwise protected with metformin.

### 3.6. Correlation between AT1 or Kidney Fibrosis Score and Biomarkers of Vascular and Kidney Injuries

To draw a link between the pathogenesis of T2DM-induced renal vascular abnormalities and kidney injury including fibrosis, we determined the correlation between AT1R or the collagen deposition score (kidney fibrosis) and the tissue and blood levels of ET-1, eNOS, iNOS, AGEs, MAP, urea and glucose. This also confirms that the role of metformin is steady and proper in serious diabetic complications such as diabetic nephropathy. Figure 4A–D shows a significant (*p* ≤ 0.0004) correlation between AT1R score and these parameters: ET-1 (r = 0.880), eNOS (r = −0.850), AGEs (r = 0.802), MAP (r = 0.957) and IL-6 (data not shown), and Figure 4E–H displays a significant (*p* < 0.0001) correlation between the collagen score and these parameters: AT1R (r = 0.899), urea (r = 0.741), iNOS (r = 0.909) and glucose (r = 0.907).

## 4. Discussion

These studies are the first to investigate renal artery expression of two signaling molecules, angiotensin II type I receptor (AT1R) and endothelin (ET-1), involved in vascular dysfunction [19] in an animal model of diabetic nephropathy. Renal artery dysfunction (stenosis) is reported in 33% of diabetic patients with hypertension and/or renal impairment, and the presence of stenosis has implications for the treatment of these patients [27]. Therefore, we investigated renal artery AT1R/ET-1/iNOS axis–mediated endothelial dysfunction, kidney injury and fibrosis in a rat model of T2DM-induced nephropathy with and without the incorporation of the antidiabetic drug metformin. This drug is thought to have pleotropic effects, including anti-oxidant properties. An association between the pathophysiology of T2DM-induced nephropathy, vascular dysfunction and glycemia was also investigated in this animal model. We therefore modeled renal artery and kidney injuries in rats secondary to T2DM and demonstrated that diabetes has the capacity to trigger renal artery AT1R/ET-1/iNOS axis–mediated endothelial dysfunction, kidney injury and fibrosis in rats 10 weeks after diabetes induction. Furthermore, this axis appeared to be inhibited by metformin (Figure 5). In addition, our data demonstrate a significant correlation between renal artery dysfunction and kidney fibrosis (collagen score) as well as these parameters also being associated with diabetes, inflammation, kidney injury and blood pressure. This demonstrates an important association between renal artery dysfunction and the development of kidney injury in diabetes as well as highlights the reno-protective effects of metformin.

Diabetic nephropathy is a common and serious microvascular complication of diabetes that, if left unchecked, results in progressive and irreversible renal failure. The maladaptive activation of the renin–angiotensin system (RAS) increases glomerular capillary pressure, thus playing a fundamental role in the pathophysiology of diabetic nephropathy progression [9,28]. Indeed, in patients with diabetic nephropathy, upregulation of angiotensin II type I receptors (AT1R) is often reported in renal biopsies [29] as well as increased urinary levels of angiotensin converting enzyme 2 (ACE2) also being observed [30]. AT1R inhibitors have been shown to ameliorate arterial blood pressure and urinary albumin excretion in T1DM patients with diabetic nephropathy [11] as well as in hypertensive patients with T2DM [31]. Both AT1R and iNOS gene and protein expression as well as inflammation and oxidative stress are increased in the renal cortex tissues obtained from diabetic rats [32], with eNOS gene polymorphism in patients with T1DM [33] and eNOS deletion in mice [13] being associated with advanced diabetic nephropathy. Metformin has been shown to decrease mean arterial blood pressure in addition to levels of AT1 and AGEs receptors in both the aorta and kidney of rats that have been fed a high-fructose diet [34]. These reports appear consistent with our data (shown in Figure 1 and Figure 2), which demonstrate an increase in MAP, a decrease in eNOS and the augmentation of AT1R, iNOS and AGEs renal artery expression in diabetic nephropathy. Furthermore, these all appear to have been ameliorated by metformin.

Diabetes and hypertension account for the majority of cases of chronic kidney disease and thus end-stage renal failure, so targeting the RAS is not enough to slow down the progression to renal failure [35]. Therefore, investigating another pathway that complements the RAS, such as ET-1, is warranted. Interestingly, Ang II can increase the expression of ET-1 [20] with the latter’s activity being upregulated in patients with type 2 diabetes mellitus (T2DM) [21]. In addition, ET-1 increases proteinuria, fibrosis and chronic kidney disease [36]. Our data demonstrate ET-1 upregulation and eNOS downregulation in renal artery tissues harvested from the model group of rats and appear consistent with a recent report showing a similar pattern of renal artery ET-1/eNOS expression obtained from obese pigs following renal denervation [37]. In addition, in vitro tissue bioassay using main branch human renal arteries demonstrated potent vasoconstriction induced by ET-1 with EC50 values of 4.06 nM [38]. Furthermore, plasma ET-1 levels have been shown to be reduced by metformin in women with polycystic ovary syndrome [39] as well as in rats with pulmonary hypertension [40]. These reports again appear to be congruous with our findings, which appear to demonstrate significant inhibition of ET-1 renal artery levels with metformin.

AT1R expression is elevated in T2DM and is positively associated with renal fibrosis. The latter, as demonstrated by the deposition of course collagen within renal sections, was observed in all diabetic rat models. AT1 receptor blockers are known to inhibit kidney fibrosis in an animal model of chronic kidney disease [25]. We have been able to demonstrate that metformin is not only able to reduce ET-1 expression but also importantly translates into reduced course collagen staining and thus fibrosis, hinting again at the pleotropic properties of metformin.

In conclusion, we demonstrated the activation of renal artery AT1R, ET-1, AGEs and iNOS as well as the downregulation of eNOS expression in a rat model of diabetic nephropathy. Metformin, a commonly prescribed, cheap and well-tolerated anti-diabetic drug is known to act in a pleotropic manner in addition to its well-recognized hypoglycemic action. We have been able to demonstrate that metformin ameliorates many of these deleterious pathways that are critical in the progression of renal artery dysfunction and chronic kidney disease, thereby providing further justification for its routine use in patients with diabetes, irrespective of glycemic control.

### Limitations of the Study

We demonstrated an association between metformin and suppression of the AT1R/ET-axis. However, to conclusively determine that protein expression is inhibited by metformin, we suggest a future study that examines the use of specific inhibitors of downstream signaling pathways.

While we demonstrated that metformin reduces diabetes-induced hypertension through modulation of renal artery iNOS and eNOS gene expression, it would have been additionally favorable to measure protein concentration as a consequence of such gene expression modulation.

The addition of profibrogenic biomarkers such as alpha smooth muscle–actin and tissue inhibitors of metalloproteinases-1 would have been complementary to the study’s finding of coarse collagen deposition within the renal interstitium and surrounding blood vessels.

Finally, SGLT2 inhibitors have recently been described as the ‘new frontier’ in diabetes care. Future studies should also interrogate whether this class of drug is also able to ameliorate renal artery AT1R/ET-1 axis–mediated kidney injury to the same extent as metformin.

## Figures and Tables

**Figure 1 biomedicines-10-01644-f001:**
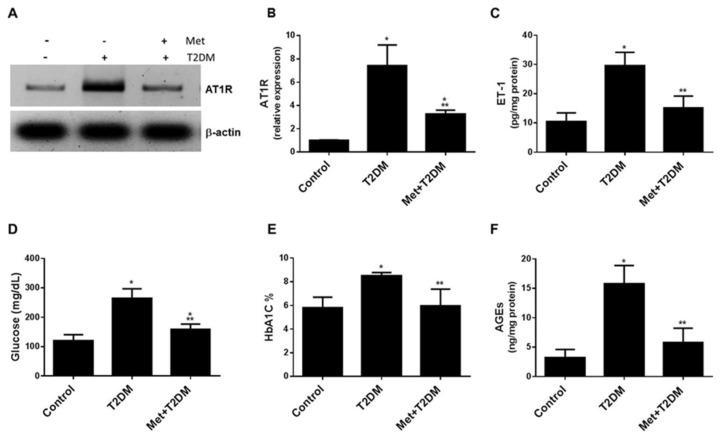
Diabetes activates the AT1R/ET-1 axis with suppression being associated with metformin. At the end of the animal experiment, Western blots of AT1R (**A**,**B**) and ELISA analysis of ET-1 (**C**) of renal artery tissue samples collected from the rat groups (Control, T2DM and Met + T2DM) are presented. The levels of glucose (**D**) and glycated hemoglobin (**E**) in the blood, as well as the levels of AGEs in the renal arteries (**F**) in the animal groups stated above, were measured at the end of the experiment. All of the *p* values shown are significant. ** *p* ≤ 0.0013 versus T2DM, * *p* < 0.03 compared control. T2DM: type 2 diabetes mellitus; AT1R: angiotensin II type I receptor; ET-1: endothelin-1; HbA1C: glycated hemoglobin; AGEs: advanced glycation end products.

**Figure 2 biomedicines-10-01644-f002:**
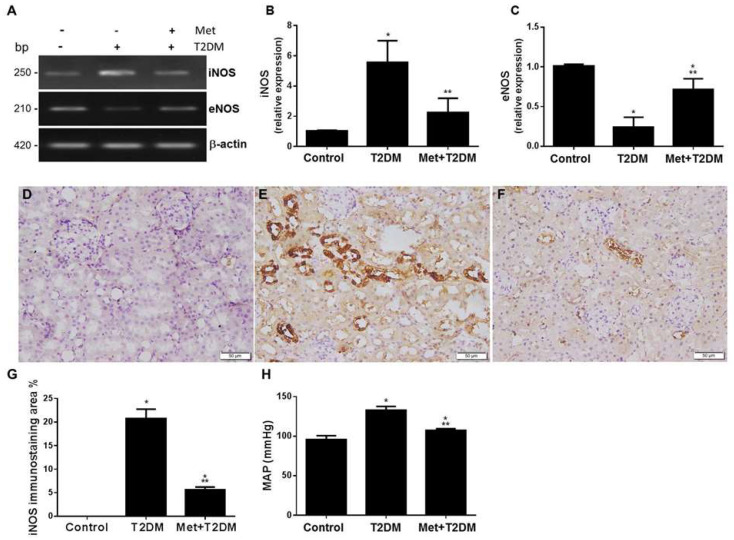
Metformin protects against diabetes-modulated NOS enzymes in renal artery and kidney tissue. Metformin (Met) reduces diabetes-induced hypertension through modulating renal artery iNOS and eNOS gene expression. At the end of the animal experiment, qRT-PCR analysis (**A**–**C**) of iNOS (**A**,**B**) and eNOS (**A**,**C**) mRNA of renal artery tissue samples acquired from the rat groups (Control, T2DM and Met + T2DM) is shown. Immunohistochemistry of iNOS (200×) of kidney sections taken from the rat groups Control (**D**), T2DM (**E**) and treatment group, Met + T2DM, (**F**,**G**) are displayed. The histograms in (**G**) show a quantitative study of the percentage of iNOS immunostaining region in kidney slices from the various groups. (**H**) Mean arterial blood pressure values were measured in all animal groups at the end of experiment. All of the *p* values shown are significant. * *p* < 0.02 versus control, ** *p* < 0.001 versus T2DM. T2DM: type 2 diabetes mellitus; iNOS: inducible nitric oxide synthase; eNOS: endothelial nitric oxide synthase; β-actin: beta-actin; MAP: mean arterial blood pressure.

**Figure 3 biomedicines-10-01644-f003:**
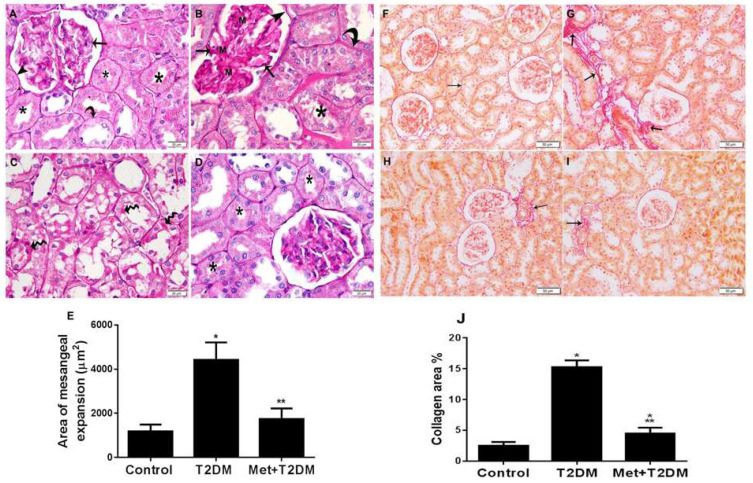
Induction of mesangial matrix expansion and kidney fibrosis by diabetes appears to be ameliorated by metformin. At the end of the animal experiment, PAS-stained pictures (400×) of kidney sections taken from the rat groups Control (**A**), T2DM (**B**,**C**) and treatment group, Met + T2DM, (**D**) are shown. The glomerular (arrow), tubular (curved arrow) and parietal layer (arrowhead) basement membranes, as well as the brush borders of tubular cells, show high positive PAS staining (star). Positive PAS staining in the tubular epithelium is indicated by wavy arrows in (**C**). M stands for the growth of the mesangial matrix. The histograms in (**E**) show a quantitative examination of mesangial growth in kidney slices from the different groups. Sirius red-stained pictures (400×) of kidney sections obtained from the rat groups Control (**F**), T2DM (**G**) and treatment group, Met + T2DM, (**H**,**I**) are shown. The arrows in (**G**) indicate strong positive staining, while the arrows in (**F**,**H**,**I**) indicate faint positive staining. The histograms in (**J**) demonstrate a quantitative percentage analysis of collagen deposition (fibrosis) in kidney slices from each of the groups All of the *p* values shown are significant. * *p* < 0.0001 versus control, ** *p* < 0.0001 compared to T2DM. T2DM: type 2 diabetes mellitus.

**Figure 4 biomedicines-10-01644-f004:**
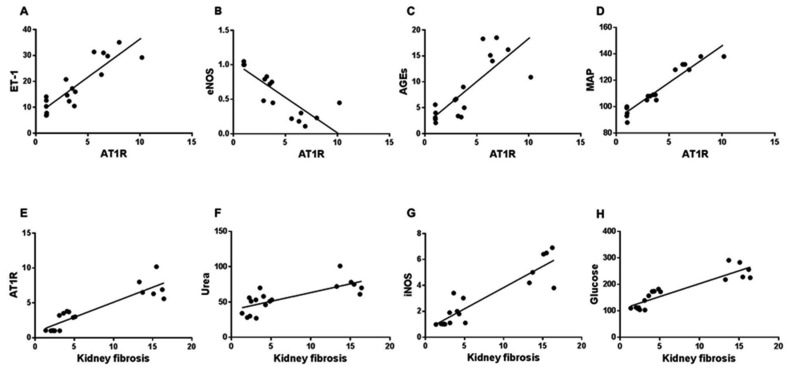
The AT1R or kidney fibrosis score correlates with biomarkers of renal artery and kidney injury as well as glycemia. All rat groups had their renal artery AT1R expression and collagen deposition in kidney tissues assessed and the relationship between either AT1R and ET-1 (**A**), eNOS (**B**), AGEs (**C**) and MAP (**D**) or kidney fibrosis versus AT1R (**E**), urea (**F**), iNOS (**G**) and glucose (**H**) are shown. AT1R: angiotensin II type I receptor; ET-1: endothelin-1; eNOS: endothelial nitric oxide synthase; AGEs; advanced glycation end products MAP: mean arterial blood pressure; iNOS: inducible nitric oxide synthase.

**Figure 5 biomedicines-10-01644-f005:**
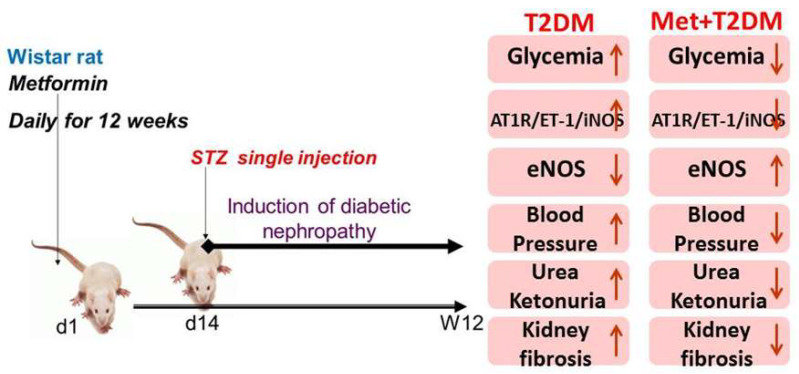
Proposed model for diabetes-induced nephropathy which appears to be inhibited by metformin. STZ: streptozotocin; T2DM: type 2 diabetes mellitus; Met: metformin; AT1R: angiotensin II type I receptor; ET-1: endothelin-1; iNOS: inducible nitric oxide synthase; eNOS: endothelial nitric oxide synthase.

**Table 1 biomedicines-10-01644-t001:** Effects of metformin (Met) on diabetes-modulated inflammation, oxidative stress and antioxidants. Blood levels of hs-CRP, TNF-α, IL-6, MDA, SOD and GPx were evaluated in all rat groups 10 weeks after diabetes induction. Values are expressed as mean ± SD for each group. Presented *p* values are all significant (*p* < 0.05). ^a^: significant in comparison to control; ^b^: significant in comparison to T2DM. T2DM: type 2 diabetes mellitus; hs-CRP: high sensitivity C-reactive protein; TNF-α: tumor necrosis factor-alpha; IL-6: interleukin-6; MDA: malonodialdehyde; SOD: superoxide dismutase; GPx: glutathione peroxidase.

Animal Groups	hs-CRP (μg/mL)	TNF-α (pg/mL)	IL-6 (pg/mL)	MDA (nmol/L)	SOD (U/L)	GPx (nmol/min/mL)
Control	1.22 ± 0.28	33.32 ± 2.26	16.60 ± 4.91	13.44 ± 2.97	2.41 ± 0.40	161.3 ± 9.07
T2DM	8.24 ± 2.10 ^a^	113.00 ± 7.93 ^a^	118.70 ± 6.85 ^a^	80.44 ± 6.42 ^a^	0.74 ± 0.160 ^a^	80.57 ± 5.62 ^a^
Met + T2DM	4.29 ± 1.31 ^ab^	55.29 ± 13.57 ^ab^	37.29 ± 10.25 ^ab^	28.09 ± 11.46 ^ab^	2.07 ± 0.52 ^b^	114.6 ± 5.29 ^ab^

**Table 2 biomedicines-10-01644-t002:** Effects of metformin (Met) on diabetes-modulated kidney injury biomarkers and body weight. Urine albumin, ketones, specific gravity and blood levels of urea and creatinine as well as body weight were evaluated in all rat groups 10 weeks after diabetes induction. Values are expressed as mean ± SD for each group. Presented *p* values are all significant (*p* < 0.05). ^a^: significant in comparison to control; ^b^: significant in comparison to T2DM. T2DM: type 2 diabetes mellitus.

Animal Groups	Urine Albumin (mg/24 h)	Ketonuria (mmol/L)	Urine Specific Gravity (pg/mL)	Urea	Creatinine (mg/dL)	Body Weight (gram)
Control	13.71 ± 3.69	0.00 ± 0.00	1005.83 ± 2.04	37.67 ± 12.60	0.17 ± 0.06	270.0 ± 5.25
T2DM	42.86 ± 4.67 ^a^	5.54 ± 2.25 ^a^	1030.0 ± 0.00 ^a^	78.75 ± 15.49 ^a^	1.05 ± 0.32 ^a^	163.3 ± 7.52 ^a^
Met + T2DM	24.57 ± 6.02 ^ab^	1.18 ± 1.64 ^b^	1020.83 ± 5.85 ^ab^	49.89 ± 11.21^b^	0.533 ± 0.23 ^ab^	305.83 ± 31.05 ^ab^

## Data Availability

The data that support the findings of this study are available on request from the corresponding author.

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
