# Peer review of "Metformin Is Associated with the Inhibition of Renal Artery AT1R/ET-1/iNOS Axis in a Rat Model of Diabetic Nephropathy with Suppression of Inflammation and Oxidative Stress and Kidney Injury"

_biomedicines, 2022, doi:10.3390/biomedicines10071644_

Round 1
Reviewer 1 Report
Overall, the manuscript only shows a correlation, it does not prove any causation. The manuscript shows that metformin suppresses AT1R and ET-1 expression and increases eNOS in renal arteries and ameliorate diabetic renal damage in histologically and serologically. Figure 6 shows the correlation AT1R and renal fibrosis with various markers, but no causal relationship has been demonstrated. To address this issue, the authors are supposed to perform additional studies with ARBs and endothelin antagonists. At least, I find the title and abstract unacceptable.
Major;
1. Fig 1 is the data that metformin suppressed AT1R and ET1 expression, not proof of inhibition of AT1R/ET-1 axis. To prove it, the author should exam the downstream signals using some inhibitors.
2. The concentration of immunostaining does not indicate the amount of protein expression. The arrows are unnecessary in Fig. 3.
3. The authors need to discuss the weight change across each group in Table 1.
Minor;
1. It is difficult to understand (B) and (D) in line 191 and (C) in line 214.
Reviewer 2 Report
The authors in the manuscript entitled 'Metformin Inhibits Renal Artery AT1R/ET-1 Axis Mediated Kidney Injury and Fibrosis Associated with the Modulation of NOS Enzymes in Diabetes-Induced Nephropathy' have shown the link between the renal artery receptor angiotensin II type I (AT1R) and endothelin-1 (ET-1) involved in vasoconstriction, oxidative stress and inflammation and kidney fibrosis (collagen) in diabetes-induced nephropathy with and without metformin administration in rats. The authors need to consider the following points to improvise the manuscript.
1. In Figure 1. Metformin (Met) inhibits the AT1R/ET-1 axis, which is activated by diabetes. Fig.1A Include ET-1 western blot and Fig 1C. include ATIR ELISA. This will add rigor to the experimental study.
2. In Figure 2. Metformin (Met) reduces diabetes-induced hypertension through modulating renal artery iNOS and eNOS gene expression. Is there any change in the protein levels with or without metformin treatment in T2DM rats?
3. Figures 2 and 3 can be combined. In Figure 3, only show one high-resolution representative image of Control, T2DM, and T2DM+metformin. Rest can be added as supplementary figures.
4. Figures 4 and 5 can be combined. Additionally, include a few fibrosis markers (qPCR).
5.The authors have mentioned that angiotensin II type I (AT1R) and endothelin-1 (ET-1) involved in vasoconstriction, oxidative stress and inflammation. The manuscript lacks data on inflammation and oxidative stress. Include inflammatory and oxidative stress markers (qPCR and staining) to show the involvement of AT1R and ET-1 in diabetes-induced nephropathy with and without metformin administration in rats.
6. What happens to the kidney injury markers NGAL and KIM-1? Are there any changes after metformin treatment?
Round 2
Reviewer 1 Report
The authors have corrected the overstatement and evaluated the data objectively.
Reviewer 2 Report
The authors have addressed the relevant comments.